# Pandemic-related experiences of older adults and people with disabilities

**W. Ben Mortenson**[1,2,5], **Elham Esfandiari**[2,3], **Somayyeh Mohammadi**[1,2],
**Brodie Sakakibara**[1,3], **Julia Schmidt**[1,2], **Ethan Simpson**[1,2,3], **Janice Chan**[1], **Holly Reid**[1,4],
**Isabelle Rash**[1,2], **Emily Brooks**[1], **Gordon Tao**[1,2], **Quinn Krahn**[1], **Jessica Irish**[1],
**Susan Forwell**[1,5], **Catherine Backman**[1], **Jaimie Borisoff**[1,5,6], **Nicole Ketter** [1], **Natalie Yu**[2],
**William C. Miller** [1,2,4]*

1 Department of Occupational Science and Occupational Therapy, University of British Columbia,
Vancouver, British Columbia, Canada, 2 GF Strong Rehabilitation Research Lab, Vancouver Coastal
Research Institute, Vancouver, British Columbia, Canada, 3 Graduate Program in Rehabilitation Sciences,
The University of British Columbia, Vancouver, British Columbia, Canada, 4 Canadian Association
of Occupational Therapy (CAOT), Ottawa, Ontario, Canada, 5 International Collaboration on Repair
Discoveries (ICORD), Vancouver, British Columbia, Canada, 6 Department of Rehabilitation Engineering
and Design, British Columbia Institute of Technology, Vancouver, British Columbia, Canada

* bill.miller@ubc.ca

doi.org/10.1371/journal.pone.0325306

UNITED KINGDOM OF GREAT BRITAIN AND
NORTHERN IRELAND

**Peer Review History:** PLOS recognizes the
benefits of transparency in the peer review
process; therefore, we enable the publication
of all of the content of peer review and
author responses alongside final, published
articles. The editorial history of this article is
available here: https://doi.org/10.1371/journal.
pone.0325306

## Abstract

The COVID-19 pandemic has had wide reaching effects especially for people with
disabilities. Drawing on Sen's Capability approach, we explored experiences of peo-
ple with disabilities and older adults using semi-structured interviews that were con-
ducted three to four months after the pandemic was declared. We recruited 71 adults
from British Columbia, Canada: adults with spinal cord injuries (n = 22), adults who
have experienced a stroke (n = 26), adults with general disabilities (n = 13), and older
adults (over the age of 65) without reported disabilities (n = 10). Our analysis identi-
fied one overarching theme: "navigating the new normal: the societal reconfiguration
of functionings and capabilities" described the challenges and successes participants
experienced during pandemic-related restrictions and regulations. This theme encap-
sulated three sub-themes: 1) "Trying to stay connected: resuming functionings via
alternative means," 2) "Worrying about future capability opportunities," and 3) "Want-
ing to exert control in the face of constrained capabilities." The pandemic created
unique challenges for people with disabilities. The capability approach appears to be
a helpful lens to interpret pandemic restrictions among people with disabilities, which
could inform future policy.

## Introduction

The COVID-19 pandemic led to the introduction of widespread changes to the way
our society is organized. Following the declaration of a global pandemic, health
authorities issued a variety of evolving orders and recommendations around issues

**Data availability statement:** In accordance with our institution's ethics policy, anonymising qualitative interview transcripts does not make them fully unidentifiable, therefore data can be requested through Borealis (https://doi.org/10.5683/SP3/FOJDLJ) by researchers who meet the criteria for access to confidential data.

**Funding:** The author(s) received no specific funding for this work.

**Competing interests:** The authors have declared that no competing interests exist.

like physical distancing, wearing masks, and handwashing. Many jurisdictions initiated partial lock downs in which some businesses were closed and restricted the number of customers at any one time, and some activities were banned or modified (e.g., indoor exercise).

Restrictions due to the pandemic have disproportionately affected those who are already marginalized (e.g., older adults and people with disabilities) by many pre-existing structural inequities related to income, race, sex, gender, age, and ability. For example, in the United Kingdom, people residing in the most economically disadvantaged areas were twice as likely to die from COVID-19 compared to those living in the most prosperous areas, which is likely related to interactions among poverty, poor housing, and pre-existing health inequities [1]. African Americans in the United States have been found to be more than three times likely to contract COVID-19 compared to White Americans due to structural inequalities that are present in systems of power [2]. In many countries (e.g., United States and Germany), women experienced more COVID-19-related job losses compared to men [1] and, because of social norms, experienced more problems with childcare related to school and daycare closures [3]. In the face of entrenched ableism, people with disabilities were concerned they may not be prioritized to receive emergency medical interventions [4]. This entrenched ableism overlays with agism where older adults' worth of life may be deemed as lesser, resulting in lower priority for care, despite death rates due to COVID-19 increasing dramatically with age. Concerns have been raised about the ability of people with disabilities to access health care services, especially if they cannot be addressed via telemedicine [5]. A survey from over 2,400 people with disabilities in the United States, a month after the pandemic was declared, found 44% experienced new challenges accessing health care and 56% had their regular health care disrupted [6], which may have negative long-term health consequences. Many of these structural inequalities are likely intersectional in nature [7] as evidenced by trajectories of disability that vary depending on race and gender for example [8]. In contrast, the increased possibility of working from home has been identified as a potential "silver lining" among workers with disabilities who can take advantage of it [9]. A Canadian survey of workers during the first wave of the pandemic found that those with mental and physical health disabilities had worse employment conditions (e.g., stress, more contract work), and reported greater financial and health concerns, than those without a disability [10].

Less is known about how older adults and people with disabilities have experienced these societal changes. An intersectional study in Northern India among three groups (widows, people with disabilities, people with disabilities living in slums) found that the latter group experienced almost no access to health care and health information and experienced the most despair, mental distress, income loss, and hunger [11]. A systematic review of the impact of quarantine related social isolation among older people acknowledged the importance of health strategies given the increased vulnerability and susceptibility of older adults, but found it had wide-ranging, negative mental and physical affects including depression, anxiety, poorer sleep quality, and reduced physical activity [12].

We conducted a longitudinal study during the first year of the COVID-19 pandemic that included four groups of individuals, adults with spinal cord injuries (SCI) (N = 22), adults who had experienced a stroke (N = 26), older adults (65 years or older) without reported disabilities (N = 10), and adults with other self-disclosed disabilities (N = 13). These groups represented marginalised populations who were potentially disproportionately impacted by restrictive measures due to factors such as higher risk of infection, limited mobility, and discrimination. Longitudinal data from participants with spinal cord injury, stroke, and participants with other disabilities identified a typology of three types of participant profiles: those who felt extremely constrained by pandemic related restrictions, those who tried to adapt and find new routines, and those who accepted the situation and found greater appreciation for life [13]. Longitudinal data from older adults without self-identified disabilities emphasized how they attempted to adapt in light of ongoing restrictions and how many developed a sense of acceptance [14]. Data from individuals with spinal cord injuries in the first few months of the pandemic emphasized the restrictions to mobility and daily life they experienced and, despite their attempts to remain resilient, the negative impacts this had on their emotional health and social participation [15]. To develop a more critical understanding of the experiences of people with disabilities and older adults at the beginning of the pandemic, this paper draws upon the capability approach as a theoretical lens [16–18]. Sen proposed an understanding of well-being that emphasizes the importance of giving people the freedom to do activities that are meaningful to them and be what they desire. Capabilities are not physical or cognitive abilities that individuals possess, but instead represent the real opportunities they have for doing and being, in consideration of the inextricable combination of an individuals' capacities, resources they have access to, and the broader social, political, and cultural environment. Thus, capabilities are the opportunities people have given their personal circumstances and context. Functionings are the capabilities that a person has achieved either deliberately or unintentionally. Functionings and capabilities depend on conversion factors, which determine how much individuals can benefit from any resources at their disposal (e.g., goods and services). There are three types of conversion factors: personal factors that are innate to the individual, social factors that are related to interpersonal interactions and societal norms and practice, and environmental factors which are related to the built/physical environment. Conversion factors reveal the distinction between resources and capabilities. For example, a resource like a car cannot function as a viable means of transportation if an individual lacks the skills to use it or the requisite licencing (personal), lives in a society which does not permit them to drive because of their age or gender (interpersonal), or lives in a place that has roads that cannot be driven on with the vehicle that they own (e.g., unpaved logging roads instead of paved streets) (environmental).

Although health is a major focus of the capability approach, both Sen and Nussbaum have considered its applications to people with disabilities and aging [19–21]. A focus on the equal distribution of capabilities addresses concerns that the equal distribution of resources results in inequity as people with disabilities have different personal conversion factors than people without disabilities [22]. Furthermore, they may experience more discrimination, lack funding to acquire necessary equipment (social conversion factors), and experience accessibility issues with the built environment (environmental conversion factors). They may also experience stigmatization as another potential negative social version factor [23]. As an application of the capability approach Mitra created the Human Development Model of Disability, Health, and Well--being, which separates impairments and health conditions from personal factors so their causes and consequences can be explicitly analyzed [24]. According to the Human Development Model of Disability, Health, and Wellbeing, disability is a loss of functioning(s) and/or capability(s) of individuals that results from how available resources, personal and structural factors, and personal health factors intersect [20]. Although the approach has been critiqued for being too individualistic [23], others have extended it to consider capabilities more collectively (e.g., in the case of families or more communal societies) [24–26]. For example, Trani et al have proposed a framework that considers how public policies create community level capability sets which determine what valued functionings should be available to all community members. Conversion factors operating at the individual, family, community, and state level then create individual capability sets [22].

With regard to a pandemic, like COVID-19, the capability approach has recently been recommended as a way to inform societal responses that reduce, rather than exacerbate, health and other inequities [27]. Drawing on Sen's capability

approach, the purpose of the study was to explore the pandemic-related experiences of older adults and people with a variety of disabilities during the early stages of the pandemic. We were interested in finding out how these individuals were impacted by pandemic-related restrictions and how they attempted to deal with them.

## Methods

Data for this study were collected as part of a larger, longitudinal mixed-methods study [28]. These qualitative interviews were conducted by videoconference at the first time point (May 2020 to June 2021). This aligned with the first phase of British Columbia's COVID-19 response that involved the declaration of a public health emergency, and subsequent implementation of a variety of measures including physical distancing requirements, a ban on mass gatherings, halting non-essential personal services, and postponing non-urgent surgeries [28]. Ethical approval was obtained from the University of British Columbia Behavioural Research Ethics Board and the Vancouver Coastal Health Research Institute (H20-01109). Prior to their initial interviews, participants emailed a signed copy of their consent form to the research team. The study has been reported according to the Consolidated Criteria for Reporting Qualitative Research [29].

### Participants and recruitment

Recruitment started in April 2020, and we recruited participants 19 years and older from the province of British Columbia (BC) from four groups: older adults (65 years or older) without reported disabilities, adults with spinal cord injuries (SCI), adults who had experienced a stroke, and adults with other self-disclosed disabilities. Participants also needed to be able to communicate in English. Potential participants with moderate or severe cognitive impairment or aphasia were excluded.

We recruited participants in three ways: 1) on-line postings on the Reach BC and International Collaboration on Repair Discovery websites; 2) emails to previous study participants who consented to being contacted about future studies (including people with stroke, brain injury, amputation, assistive technology users); and 3) via social media posts. We screened potential participants to determine eligibility and reviewed the written study consent form with them. Prior to their initial interviews, participants emailed a signed copy of their consent form to the research team.

Participants are described in Table 1 below. Those with SCI had the lowest mean age. The majority of participants were born in Canada. Older adults were predominantly female whereas people with stroke were predominantly male. Most participants were college or university educated. Participants with stroke were least likely to live alone, in contrast to those with other disabilities. All participants in the older adult group were retired as were a majority of participants with stroke. Given the heterogeneity of disability, the "people with other disabilities group" included individuals with an array of disabilities, including physical (cerebral palsy, arthrogryposis multiplex congenita, Ehlers-Danlos syndrome, arthritis, multiple sclerosis, amputation, post-polio syndrome), mental (depression, anxiety), sensory (vision impairment) and cognitive (brain injury).

### Data collection and analysis

Before each interview, we used an online survey tool (Qualtrics) to collect demographic information about each participant (i.e., age, sex, gender, education level, living situation, income, country of birth, employment). Please see our protocol paper for details about the demographic survey [28]. We asked participants if they had been exposed to someone with COVID-19 or tested positive themselves. Using an interview guide, one-to-one semi-structured interviews were conducted and recorded using online video conferencing (i.e., Zoom). The interview guide explored changes related to the pandemic, such as alterations to daily activities, experiences with COVID-19 precautions, social support and interactions, and coping strategies. A complete copy of the interview guide is available in our protocol paper [28]. Participants received a $30 (CAD) honorarium. Eleven trained individuals conducted the interviews. Interviewers were current graduate students or practicing health care professionals. They varied in terms of gender, country of origin, and race. All were younger than 34 years of age, seven were female, one was trans, two disclosed having a disability. Teams of two or three interviewers conducted all the interviews for each of the four participant groups.

**Table 1. Participant demographics.**

| | | Older adults (OA) | People with SCI (SCI) | People with stroke (CVA) | People with other disabilities (PWOD) |
|---|---|---|---|---|---|
| Number | | 10 | 22 | 26 | 13 |
| Age | Mean (SD) | 72.60 (4.19) | 53.77 (11.06) | 65.5 (12.7) | 55.76 (16.72) |
| Sex | Male | 3 (30%) | 12 (54.5%) | 18 (69.2%) | 5 (38.5%) |
| | Female | 7 (70%) | 9 (40.9%) | 7 (26.9%) | 8 (61.5%) |
| | Not disclosed | – | 1 (4.5%) | 1 (3.8%) | – |
| Gender | Man | 3 (30%) | 12 (54.5%) | 19 (73.1%) | 5 (38.5%) |
| | Woman | 7 (70%) | 9 (40.9%) | 7 (26.9%) | 6 (46.2.5%) |
| | Other (e.g., not disclosed; two-spirit) | – | 1 (4.5%) | 1 (3.8%)* | 2 (15.3%) |
| Education Level | Some high school | – | – | 1 (3.8%) | – |
| | Graduated from high school | 1 (10%) | 1 (4.5%) | 2 (7.7%) | – |
| | Some trade school | – | | 1 (3.8%) | – |
| | Graduated trade school | – | 1 (4.5%) | 2 (7.7%) | – |
| | Some college or university | 3 (30%) | 5 (22.7) | 8 (30.8%) | 1 (7.7%) |
| | Graduated from college or university | 4 (40%) | 12 (54.5%) | 6 (23.1%) | 7 (53.8%) |
| | Some post graduate school (e.g., PhD) | – | 1 (4.5%) | 2 (7.7%) | 1 (7.7%) |
| | Graduated from post graduate school (e.g., PhD) | 1 (10%) | 2 (9.1%) | 1 (3.8%) | 3 (23.1%) |
| | Other | 1 (10%) | – | 3 (11.5%) | 1(7.7%) |
| Living Situation (participants could select more than one option) | Living alone | 5 (50%) | 7 (31.8%) | 7 (26.9%) | 9 (69.2%) |
| | Living with a partner | 4 (40%) | 8 (36.4%) | 13 (50%) | 2 (15.4%) |
| | Live with one or more of your children | 1 (10%) | 1 (4.5%) | 2 (7.7%) | 1 (7.7%) |
| | Live with a different family member | 0 | 3 (13.6%) | 1 (3.8%) | 2 (15.4%) |
| | Live with a friend or roommate | 0 | 1 (4.5%) | 0 | 0 |
| | Living with someone else | 0 | 1 (4.5%) | 6 (23.1%) | 1 (7.7%) |
| | Living in retirement/assisted living | – | 1 (4.5%) | 1 (3.8%) | – |
| Income | Less than $14,999 | – | 2 (9.1%) | 2 (7.7%) | 1 (7.7%) |
| | $15,000 to $44,999 | 2 (20%) | 5 (22.7%) | 11 (42.3%) | 7 (53.8%) |
| | $45,000 to $74,999 | 3 (30%) | 6 (27.3%) | 6 (23.1%) | 2 (15.4%) |
| | Greater than $75,000 | 4 (40%) | 3 (13.6%) | 7 (26.9%) | 3 (23.1%) |
| | Not disclosed | 1 (10%) | 6 (27.3%) | – | – |
| Country of Birth | Canada | 5 (50%) | 14 (63.6%) | 15 (57.7%) | 12 (92.3%) |
| | Others | 5 (50%) | 7 (36.4%) | 11 (42.3%) | 1 (7.7%) |
| Employment | Employed full-time | – | 3 (13.6%) | 1 (3.8%) | 1 (7.7%) |
| | Employed part-time | | 3 (13.6%) | 2 (7.7%) | 1 (7.7%) |
| | Student | – | – | 1 (3.8%) | 1 (7.7%) |
| | Retired | 10 (100%) | 5 (22.7%) | 15 (57.7%) | 4 (30.8%) |
| | On a disability assistance | – | 7 (31.8%) | 2 (7.7%) | 4(30.8%) |
| | Housemaker and Other (e.g., Independent Contractor, etc.) | – | 3 (13.6%) | 3 (11.5%) | 2 (15.4%) |
| | Unemployed | – | 1 (4.5%) | 2 (7.7%) | – |

*one participant identified as more than one gender.

Interviewers and additional team members conducted a thematic analysis of the data. First, we drew upon elements of Braun and Clarke's [30] approach to reflexive thematic analysis, namely their 5 steps: (I) familiarization with the data, 2) development of initial codes, 3) identification of preliminary themes, 4) review of preliminary themes, and 5) designation

of final themes. Then, because we had multiple analysts, we created a coding guide. To address concerns identified by Braun and Clarke [30,31] that the process should be organic, we did not use *a priori* codes and allowed our coding guides to evolve as more interviews were analyzed and through discussion between primary analysts and the full research team. To do so, the coders from each group of population coded the first three interviews separately and then compared the codes to develop the first draft of the coding guide (i.e., independent analyses were first conducted for each sub-sample). Sub-sample coding guides were then compared with other team members (the first author supervised the development of the coding guide for each group) and the entire team collaborated to create a cross-sample coding guide that was inclusive of the diverse perspectives of each team member involved (i.e., complementary rather than consensus). This preliminary coding guide was then applied to subsequent interviews and revised in consultation with the qualitative research team for each participant group as the analysis progressed when new codes were identified, or older codes needed to be reconfigured. After coding all interviews, within each participant group, codes were clustered into potential categories. Through research team discussions, categories were defined, debated, and regrouped until the researchers were satisfied they represented the data set. We then pooled the data across all participant groups and combined categories. Categories were grouped into themes and linkages among themes identified. We used the capability approach as sensitizing concept which informed, but did not predetermine, our analysis.

We used three main trustworthiness strategies: reflexivity, inclusion of multiple interviewers, and inclusion of multiple researchers in the analysis process. To promote reflexivity, interviewers kept a research diary to record their biases, assumptions, and emerging understandings and recorded self-reflections after each interview. Details about the positioning of the interviewers is provided above. The inclusion of multiple researchers with diverse positioning during data collection and analysis provided rich complementary perspectives.

## Results

Our analysis identified one overarching theme, "*navigating the new normal: the societal reconfiguration of functionings and capabilities,*" that encapsulated three subthemes that reflected how participants responded to living during the early months of the pandemic: 1) "*trying to stay connected: resuming functionings via alternative means,*" 2) "*worrying about the future capability opportunities,*" and 3) "*wanting to exert control in the face of constrained capabilities.*" We used group designations and participant numbers rather than pseudonyms to protect their anonymity.

### Navigating the new normal: The societal reconfiguration of functionings and capabilities

This theme a) revealed how participants frequently struggled to follow pandemic related recommendations because of their health conditions, b) identified issues related to the built and social environment in relation to public health recommendations, c) examined how restrictions restricted their freedoms and functionings including access to health-care, and d) explored how they attempted to adapt.

The COVID-19 pandemic led to the introduction of an evolving series of public health recommendations and regulations that restricted participants capabilities and demanded modifications to functioning, which included advice about handwashing, changing recommendations regarding mask wearing, calls for physical distancing, and service changes (e.g., closure of some businesses, reducing the number of patrons, move toward virtual services). For example, a 76-year-old female (OA 2) indicated: "I find it frustrating not being able to […] go anywhere […] without thinking, have I got the mask, […] the poles, […] the hand sanitizer […]. It's a bit like carrying an extra load in life, this pandemic thing".

Because of their health conditions, some participants struggled to follow these recommendations. A 70-year-old female with stroke noted: "I had to wear the mask when we went on the bus and I found it awfully uncomfortable and hard to breathe through. First with [my] hands. It's not something I can do anyway because I can't move this other hand." (CVA 1) A 53-year-old male with stroke (CVA 15) reported difficulties using his wheelchair on transit initially: "Before I would usually

get parked on the right side of the bus, the driver would have to clamp me in and put hooks on me, but now they're not allowed to do that. […] if I didn't have a disability, I could just get on the bus like everyone else." So in this way their functionings were constrained by change in environmental conversion factors.

Participants also encountered challenges dealing with others in the community regarding COVID-19 restrictions. A 50-year-old female with quadriplegia (SCI 12) described how physical distancing was more challenging for wheelchair users: "I think it's probably more difficult for me than for an abled body person just because I take up more space and there's only certain areas I can move to. You know, I can't, um, dodge or veer as well as somebody who's walking. And also, there's just places in stores or on the street, like I can't just jump off the curb if somebody wants to walk by me." Participants also struggled with those who did not follow the current precautions as noted by a 71-year-old male with incomplete paraplegia (SCI 24).

> I was on my scooter in an elevator in the grocery store, and there were signs everywhere about social distance. And I'm in the elevator waiting for the door to close to go up to the actual grocery store, and this woman comes charging in and stays and I said, get out and she goes, I'm healthy, I said. But I'm not, but it's like I mean, I was wearing my mask when she wasn't.

In these cases, the capabilities of able bodies individuals appeared to restrict the capabilities of those with disabilities.

For most participants, COVID-19 restrictions dramatically altered their capabilities and functionings (e.g., exercise, shopping, employment). A 76-year-old female (OA 2) did not feel the same motivation to exercise given the current restrictions: "[…] I can be motivated to get to the gym or to the rec center and join the people there. Whereas I don't get it don't get motivated to do [exercise] at home." Shopping was a challenge for many participants. "Like you can't even go to a grocery store on the weekends because it's so busy. Nobody's wearing masks, nothing. And then you look like a weirdo when you wear your mask because everybody is staring at you, wondering why you're wearing a mask. It's frustrating" (SCI 6, 40-year-old female). Employment and volunteer work was a concern for participants who were not retired. "I guess one of the biggest challenges is my job, […] my job had to shut down. […] That is a big challenge." (CVA 18, 51-year-old female). Participants who were on disability benefits also experienced loss of income in some cases, as described by a 46-year-old male with incomplete tetraplegia (SCI 2). "I score keep for sports tournaments, but they all got cancelled. And because my money is cash, I get no [relief funding] or anything." Not all participants experienced financial affects with the pandemic. For example, an 80-year-old female (OA 7) revealed:

> The pandemic didn't financially impact me that much because I was already retired. […] In fact, I'm saving money because I'm shopping less […]. There's no such thing as an impulse buy anymore, because I'm not going into stores. I'm not paying for movies or lunches out or dinners out. So in some ways it's been financially better.

In this regard financial difficulties or savings were not universal among participants but could have a powerful impact on their capabilities and functionings.

Some participants had experiences of being treated paternalistically by family members, (i.e., a negative structural conversion factor). A 73-year-old female (OA 6):

> My daughter wanted to get groceries for me so I wouldn't have to go to the grocery store. […] I had asked her to buy me some tomato plants. She couldn't find any. And I thought, well, this is unfair. I mean, she's not a gardener. I want more than just tomato plants. I need topsoil and manure and all this stuff. So I said, "I'm going to go out to the garden centre and if I'm doing that, I may as well get my groceries too." So, she gave me another lesson on how to put on a mask properly and take off your gloves properly and the hand sanitizer. And she just said, "Okay, don't go to more than one grocery store at a time."

Participants experienced changes with their health care. For many participants this was negative and represented a potential threat to their current and future functionings. A 40-year-old female with depression and arthritis (PWOD 6): "[I am] not having access to the medical treatments that I usually do, so [I am] having to rely on myself a lot more for the mental healthcare than I usually do." In contrast, some participants found increased access to virtual care was beneficial. A 52-year-old male with cerebral palsy (PWOD 20): "A lot of people with disabilities, have been asking for various kinds of virtual care for years just because it's so much easier than having to arrange for and to travel to get to a doctor's office." Some participants had difficulty retaining private attendants for personal care and ensuring they had the necessary personal protective equipment. A 66-year-old male with paraplegia (SCI 4) explained, "We have needs and if those aren't met we can get significantly ill. […] if we don't get caregivers in and we get pressure sores, and then we end up in the hospital with pressure sores."

Not only did participants need to change long-standing habits of functioning, but they also needed to develop new habits, like trying to stay informed without getting overwhelmed. A male stroke survivor (CVA 20) 72-years-old described how he was trying to identify fake-news: "I try to filter out um, the complete garbage versus the real fact. So, if I have a believable and verifiable source then I would discount the other nonsense."

Some participants described needing to find ways to spend their time, in light of the activities that were no longer available or difficult to access (i.e., a deprivation of functionings). A lack of activities could cause boredom as described by an 82-year-old male (CVA 13) "There's not that many options as to what I can do, so I'm a little bit bored that way." However, he felt lucky to have his family around:

> So compared to a lot of other people, I'm very fortunate. I have a lot of variety, our children all have nice, big homes, so we can go and visit them if we need to, […] So having a big family makes a huge difference. People that don't have families, it's very, very restrictive. They feel very lonely and isolated.

A 60-year-old male participant (CVA 17) replaced his dinner parties with driveway parties: "We've just gone to somebody's house social distancing outside, like a driveway party they call it, where you just sit in the driveway eight feet apart from each other and visit." Some participants identified challenges getting back into some previous activities because of COVID-19 related shortages as described by a 64-year-old female with disability related to mental health (PWOD 12): "Well, I can't find a bike because they're all sold out. Because as soon as COVID hit, everybody went out to buy a bike. Apparently, everyone but me!"

In contrast, several participants described how, because of conversion factors, COVID-19 restrictions had not had a major effect on their functionings. Because of mobility restrictions due to her recent stroke, a 68-year-old female (CVA 2) noted, COVID is more of a restriction for other people: "When I get healthy enough, when I can start running around, instead of just walking around the building, and I can't leave the property, or I shouldn't leave the property. Um, it'll probably have more of an effect [on me]."

A 73-year-old female participant (OA 6) indicated, because of her life situation, the pandemic restrictions were not that impactful:

> Well, it sounds terrible, but I don't really think I have had any big challenges. […]. Nothing major for me. I don't have elderly relatives that I'm responsible for. Um, I don't have children who have been out of work, still fully employed. And, uh, you know, I live in a single dwelling house. I've got my garden to keep me occupied, you know, so I can go all in my backyard, not stuck in a tiny apartment. And I'm just very, very fortunate. And I realize that I mean, I know it's horrible for lot of people.

Finally, some participants already had relatively secluded lives as illustrated by a 79-year-old male with stroke (CVA 7): "I consider myself laid back and was not involved doing much before COVID came. [So] there really isn't a great change for me."

**Trying to stay connected: Resuming functionings via alternative means**

As part of this theme participants predominantly described concerns about being separated from others, and the challenges they experienced implementing alternative strategies to try and restore social functionings. Most participants described experiencing an increased sense of isolation following the pandemic restrictions, which negatively impacted their well-being.

> I guess after you realize that it is going be like this for a while, then you start missing your family. I live in a beautiful place, so I have beautiful scenery out my window all the time, But, um, [living alone] there's no replacement for having a friend that you can go for dinner with or, um, go just phone and say, let's go for a walk or, um, and the hugs that you get from other people. The thing […] I have been missing the most is my personal social contact. (CVA 24, 76-year-old female).

They especially missed the opportunity for in-person interactions with people they did not live with because of COVID-19 restrictions which sought to reduce the spread of the disease (supporting the capability of living a physically healthy life at the potential cost of mental health). A 65-year-old female (OA 4): "I love to go to movies. I love to hear live music. I love to, you know, have suppers with friends. All those things totally disappeared." Given this sense of isolation, participants enacted a variety of strategies to increase social connection.

> So as far as, you know, social stuff outside of the house – even seeing family –we've kinda been doing the distancing, so other than a few visits in the car, stopping by, and doing that kind of visits, really haven't seen family either, to be honest. It's mostly Zoom calls, FaceTime stuff, and text – text stuff too. Only, again, like I said the last week or so I've started venturing out and kinda going for some distanced walks just around the neighbourhood with, with another friend who's not working at the time as well. (SCI 1, 58-year-old female)

Emphasizing the personal nature of conversation factors, some participants identified positive outcome associated with COVID-19 restrictions. An 81-year-old male with cerebral palsy (PWOD 5) indicated:

> I think I spend more time talking to my friends. And talking to my wife … and that's been positive. Because we don't pretend that we have busy things to do, the way we did, which would sometimes shorten your interactions with people. And now we don't have to do that. So, it's good.

For those with the means and abilities (a positive feedback loop between functionings and conversion factors), many saw technology as a potential means to address their isolation. A 70-year-old female (OA 8) reported: "I do Zoom once every Tuesday morning, I do a Zoom meeting either with the neighborhood walking ladies or with the Girl Guide ladies that I [used to work with]." However, online video conferencing was not an option for all participants. A 73-year-old female (OA 19) noted: "I'm on the phone all the time. […] My long-distance bill is skyrocketing because most of my buddies are in the east coast. […] I would be doing Zoom, but I don't have that capacity on my computer." Others described a sense of Zoom fatigue.

> A 51-year-old male with cerebral palsy (PWOD 20): I know some people have been having things like Zoom watch parties and different things like that. But because, because I get so zoomed in my life [Laughs.] in my professional life, in so many ways, I haven't been quick to engage zoom for social purposes. Although I guess if there were, if there were compelling enough reason to do it, certainly I would do that. Uh, that just hasn't happened yet.

Regarding the collective nature of capabilities and functionings, the COVID-19 restrictions tended to cause an increase in interactions for people living in the same home, which could be either positive or negative depending on the previous

nature of that relation. For a 58-year-old male stroke participant (CVA 8), the kids coming home from university during COVID-19 creates tension in the household:

> We were used to a certain rhythm to the day to what the house looks like. [Now] we're running the dishwasher every two days. We're doing laundry every two days. […] Our kids are adults 22 and 20 and they're used to doing whatever at the places they were at.

Some participants appreciated the opportunity to spend more time with their families. A 44-year-old male (SCI 11) revealed the benefits of being home with his children:

> The little just daily little things that kids come up with. And kids do. Predicaments they get themselves in, just seeing the seeing the world through their eyes, kinda thing. That's the coolest thing. When you're working full time in a company, not home, you don't see it. To us, it was a no brainer that I stay at home.

### Worrying about future capability opportunities

In this theme, most participants revealed the fears they had about contracting the virus and about its impact on society more broadly. Most participants reported anxiety about the COVID-19 virus in terms of the possibility of contracting it themselves or those close to them contracting it. A 60-year-old male participant with stroke (CVA 17) emphasized:

> It's said we're in this together, but we're not in the same boat because I'm not in the same boat as you, right. You're younger. You're healthier. All that sort of stuff. Yes. We have to fight this together, but you and I are in a different boat. You've got a nice big boat. Mine's a little boat with holes. [Laughs.] Don't make waves.

Many participants also identified concerns with the impact of COVID-19 more broadly (i.e., societal conversion factors): "I think the social effects of the pandemic are extremely negative. It kind of takes away from our humanity […] when you cover up your face, it's kinda like uh another barrier to communication" (A 64-year-old male who had a stroke). For many participants, the pandemic highlighted pre-existing ablest tendencies, as revealed by A 35-year-old non-binary participant with Ehlers-Danlos syndrome (PWOD 16):

> It has been a bit of a punch in the gut is the extent to which it was always possible to completely reorganize society in space and make it accessible. And that the reason why it hadn't been done is really a matter of values and whose body minds are at stake.

In this case, discriminatory practices against people with disabilities are emphasized as conversion factors that greatly restrict the capabilities of this population. Some participants were left feeling dis-interested or unmotivated ("meh"): "I've always said, '[…] If only I had time I would do this, or I would do that.' And so, I'm sitting here thinking, 'Well, I have the time, but I don't feel like doing anything.'" a 74-year-old female with post-polio (PWOD 19) reveals.

In contrast, a few participants, like A 75-year-old male (OA 3) reported feeling content in the face of the pandemic, because of the beneficial conversion factors that they possessed: "Yeah we feel very fortunate. We have a place to live that's secure and safe, and we don't have a shortage of money to buy groceries, so yeah, yeah we're doing real good."

### Wanting to exert control in the face of constrained capabilities

In this theme, participants primarily emphasized how COVID-19 restrictions had curtailed their freedoms and produced negative side effects, which impelled them to try to restore a sense of autonomy. Almost all participants described feeling

a sense of restriction in the wake of COVID-19 related societal changes. A 76-year-old female (OA 2) indicated: "I find it frustrating not being able to get out as much. […] I feel as though I have got concrete blocks on my feet because, you know, you can't go anywhere." Some, like a 70-year-old female (OA 8) reported sleep problems: "I'm not sleeping very well […] I'm just thinking about things […] you know, I find that sometimes you think about what's gonna happen in the future." Others like a 65-year-old stroke survivor described developing habits that they considered negative: "I have noticed I am drinking more during [COVID]. It's um, not to excess or every night but I have certainly had a few more drinks than normal […]. I think that has been the biggest problem for people in social isolation."

Most participants described how they attempted to adapt to these changes in a way that allowed them to be physically and mentally active to maintain control of their life and retain a degree of agency. A 68-year-old female (OA 15) indicated: "They closed the big parks. That was a problem, because we have a very nice provincial park around here. So I couldn't go there. So, I just walked around the neighbourhood." Participants identified a wide variety of strategies they used to cope with COVID-19 related societal changes. Some, like A 75-year-old male (OA 3), changed how they shopped: "I was reluctant to go to the liquor store so I phoned, or phoned or emailed orders for beer and wine." Some reported that COVID-19 restrictions provided them an opportunity to improve their health. For example, a 60-year-old male from the stroke group (CVA 17) reported: "I'm able to control my exercise, my daily routine, my food intake um, because there's no other influencers."

Many participants attributed their resilience to skills they developed to negotiate previous struggles in their lives (i.e., a positive personal conversion factor). A 53-year-old female (SCI 14) described using past experiences to help cope:

> Yeah so I just I just knew that you have to have meaning and purpose in your life and stay connected. Ten years of cognitive behavioral therapy taught me a lot. Taught me how to be resilient. And so, yeah. But more a lot of times, the more you been through in life, the more resilient you could be coming through the other end.

Despite the restrictions imposed upon them, some participants chose to express gratitude towards the frontline workers in healthcare and decision makers, as a 76-year-old female (OA 2) indicated:

> I think the Canadian, um, people, political and medical people have done the best anybody can do. […] I think they have done a superb job of educating people, and they have given a quiet, thoughtful, they've shared their personal values, they've been, the humanity is just come right across the screen.

## Discussion

Our study of the pandemic-related experiences of people with disabilities and older adults identified one overarching theme and three sub-themes. These interrelated sub-themes emphasize how participants reacted to the threat of the pandemic (i.e., worrying about the future) and enacted various strategies to deal with these uncertainties and pandemic related restrictions that were put in place to reduce the spread of the virus, but inadvertently increased feelings of social isolation (i.e., needing to exert control; trying to stay connected).

The COVID-19 pandemic has dramatically affected the capabilities and functionings of older adults and people with disabilities. The findings are congruent with Trani et al.'s policy framework [22], as the restrictions imposed represented massive curtailment of the capability set of all citizens. And yet, in practice there seemed to be wide variety in terms of how restrictions affected the functionings of older adults and people with disabilities. Instead, it appears that those who are most disadvantaged in terms of personal conversion factors such as impairments and income were more strongly affected by new norms and regulations that were implemented without their consultation and often did not take their needs into consideration [32]. For example, in the city of Vancouver, cars were banned in the largest park to allow people to visit the park more safely on bicycle or handcycles, but this excluded many disabled people from visiting if

they previously relied on their vehicles for those visits [33]. This kind of spatial exclusion may create a form of design apartheid [34] and may make people with disabilities feel out of place by being excluded from spaces where they are not wanted [35]. In this regard, the built and social environments are closely related. This was also evidenced by the social isolation experienced by study participants, which has been reported among other populations with disabilities during pandemic lock-downs in other jurisdictions [36]. Although some might argue these individuals are choosing not to do some social activities (i.e., these activities are still capabilities) it appears that many feel they are unable to do what they want to do because of the actions of others (e.g., not following precautions) and have genuine concerns about contracting COVID-19, so their real freedoms are reduced. In this way, the collective nature of capabilities is evident, in that the functionings of some individuals may negatively impact the capabilities of others. In this regard, COVID-19 restrictions represented changes for capabilities and functionings for many individuals in society, but these restrictions were not equitably distributed as those who are more privileged experienced less of a decline in capabilities and functionings. The challenges that people from marginalized groups encounter complying with COVID-19 restrictions may contribute to stigmatization and isolation [37].

The COVID-19 restrictions exacerbated previously existing inequities experienced by people with disabilities. For example, being able to maintain physical distancing if desired is a capability that was not previously available to many people with disabilities and the pandemic restrictions made this even more challenging. In this regard, built environments have traditionally provided preferential access to those who ambulate and move without devices, and as it turns out, they are also very poorly designed to enable mandated physically distancing (e.g., the minimum recommended width of sidewalks is 1.2 meters) [38]. Distancing issues are exacerbated by accessibility challenges that people with disabilities always encountered (e.g., sandwich boards and shop displays that encroach on sidewalks; limited or absent curb cuts; prohibitions against using mobility devices on the streets) [39]. Those who rely on paid personal care or need to attend in-person healthcare visits are unable to avoid contact with people outside of their homes. COVID-19 restrictions dramatically altered how many participants received and provided care to others. Participants who relied on private attendants for personal care through the Choice in Supports for Independent Living program were especially hard hit by the pandemic, because they had difficulties retaining staff and could not afford to increase their wages. Thus, many older adults and people with disabilities appear to be doubly disadvantaged as they are less able to physically distance and potentially more susceptible to COVID-19. The COVID-19 restrictions have also made it more difficult for older adults and people with disabilities to remove themselves from living situations with ongoing negative interactions, which may exacerbate domestic violence and abuse during the pandemic [40].

The findings highlight the existence of a negative feedback loop, as those who were already disadvantaged prior to COVID-19 [41], bore the brunt of the restrictions, and this made their lives even more precarious during the pandemic. For example, the sleep problems, increased worry, and reduced physical activity reported by participants in our study, which have been reported among older adults because of COVID-19 restrictions, may be self-perpetuating [42]. Some participants increased substance use following the initial lockdown of the pandemic. This underscores how not all functionings may be perceived as universally positive [43]. The capability approach has been used to emphasize how substance use results from social inequalities and practices that put people at risk of harm (i.e., structural violence) [44]. Furthermore, reduced access to healthcare likely reduced participants' ability to manage their primary and secondary health conditions, which could potentially lead to further complications and decreased function. In contrast, those who viewed themselves as relatively advantaged reported less changes in their functionings as their incomes remained relatively stable and, because of their technical skills and access to technology, they were able to take advantage of online delivery for many goods and services and various communication platforms to maintain social connections. This demonstrates the dynamic and interdependent nature of capabilities and functionings. Although functionings are normally considered to be enacted capabilities, there is a reciprocal relationship in that abilities can change as a result of functionings (e.g., improving with practice and diminishing with disuse or deconditioning) which ultimately affects capabilities.

In some cases, the precautions may have provided some unintended benefits for people with disabilities in terms of their ability to work from home, and reduced need to commute. Furthermore, in terms of conversion factors some people with disabilities may have increased resilience because of the skills they developed grappling with previous life changes [12,13,45]. That said, resilience is likely more common among those who occupy more privileged identity positions. Resilience may explain why a few participants reported feeling content in the face of the pandemic. Conversely, it may also represent a situation in which people have adjusted to their deprivations by adjusting their aspirations downwards [46]. It is also important to stress that participants continued to exercise autonomy in light of the capabilities that were available to them and were frequently proactive and creative in the strategies they implemented to deal with pandemic related restrictions in a way that allowed many of them to remain physically and mentally active. This was especially true for older participants without disabilities who are affluent. This suggests the need to consider how capabilities should be supported on a community level to enable everyone to have the same opportunities [22].

## Limitations

In terms of transferability, is must be noted the study was conducted in a province that had its own specific restrictions and that these data were collected during the beginning of the COVID-19 pandemic in a jurisdiction with relatively few cases. The findings would likely have been different in a setting with high community spread. Although we collected data about if people were born outside of Canada, we did not ask participants for racial or ethnic information. The study involves a large number of investigators and interviewers, which is unusual in qualitative research, but we tried to manage this via ongoing training of interviewers and repeated team discussions. Finally, the electronic means of recruitment and use of online video conferencing as a means of data collection excluded those without computer and internet access.

## Future research directions

To understand better how people's experiences of COVID-19 restrictions are affected by a broader range of sociodemographic factors it would be beneficial to conduct an analysis that could explore how factors like race, sex, and gender may have intersected in terms of individuals' pandemic-related experiences. Future work could focus on implementing and evaluating policy changes that promote a more disability-inclusive response to future pandemics and crisis.

## Conclusion

We used Sen's capability approach to explore the experiences of COVID-19 related restrictions among 71 participants from four groups: people with SCI, people who have experienced a stroke, people with other disabilities, and older adults without reported disabilities. Constraints to reduce the spread of COVID-19 may have disproportionately affected older adults and people with disabilities, especially those who were already disadvantaged prior to the pandemic. The findings emphasize the need for greater consultation and consideration of the needs of these populations to reduce the negative unintended consequences associated with efforts intended to deal with large scale crises like pandemics and natural disasters. This is critical to reduce inequity, maintain capabilities, and promote functionings.

## Author contributions

**Conceptualization:** W. Ben Mortenson, Elham Esfandiari, Brodie Sakakibara, Julia Schmidt, Holly Reid, Susan Forwell, Catherine Backman, Jaimie Borisoff, William C. Miller.

**Data curation:** W. Ben Mortenson, Elham Esfandiari, Somayyeh Mohammadi, Brodie Sakakibara, Julia Schmidt, Ethan Simpson, Isabelle Rash, Emily Brooks, William C. Miller.

**Formal analysis:** W. Ben Mortenson, Somayyeh Mohammadi, Brodie Sakakibara, Ethan Simpson, Janice Chan, Holly Reid, Isabelle Rash, Emily Brooks, Gordon Tao, Quinn Krahn, Jessica Irish, Susan Forwell, Jaimie Borisoff, Nicole Ketter, Natalie Yu, William C. Miller.

**Funding acquisition:** W. Ben Mortenson, Jaimie Borisoff.

**Investigation:** W. Ben Mortenson, Isabelle Rash.

**Methodology:** W. Ben Mortenson, Brodie Sakakibara, Julia Schmidt.

**Project administration:** W. Ben Mortenson, Elham Esfandiari, Somayyeh Mohammadi, Ethan Simpson, Holly Reid, Isabelle Rash, Emily Brooks.

**Supervision:** W. Ben Mortenson, Brodie Sakakibara, Julia Schmidt, Catherine Backman, Jaimie Borisoff.

**Validation:** W. Ben Mortenson.

**Writing – original draft:** W. Ben Mortenson, Elham Esfandiari, Somayyeh Mohammadi, Brodie Sakakibara, Julia Schmidt, Ethan Simpson, Janice Chan, Holly Reid, Isabelle Rash, Emily Brooks, Gordon Tao, Quinn Krahn, Jessica Irish, Susan Forwell, Catherine Backman, Jaimie Borisoff, Nicole Ketter, Natalie Yu.

**Writing – review & editing:** W. Ben Mortenson, Ethan Simpson.

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
