## [Decision Letter · Decision Letter 0]

16 Mar 2025

Dear Dr. Miller,

Thank you for submitting your manuscript to PLOS ONE. After careful consideration, we feel that it has merit but does not fully meet PLOS ONE’s publication criteria as it currently stands. Therefore, we invite you to submit a revised version of the manuscript that addresses the points raised during the review process.

Thank-you for your patience with the review process. I think there are some final revisions to take into account to ensure theoretical and methodological clarity of your work. Both reviewers have made good suggestions and reviewer two did a very thorough read and picks up on the need to describe why these populations were chosen and some clarification of the sampling design. You do need to make a stronger case for the inclusion of the capabilities approach and justify that more. There is also a need to proofread which is normal and we hope you find their suggestions useful.

We look forward to receiving your revised manuscript.

Kind regards,

Maria Berghs, PhD

Academic Editor

PLOS ONE

Journal Requirements:

“This study did not receive any funding for the research, but it was supported by Dr. Ben Mortenson’s New Investigator Award from the Canadian Institutes of Health Research.”

4. In the online submission form, you indicated that [Anonymising qualitative interview transcripts does not make them fully unidentifiable, therefore data can be requested through Borealis (https://doi.org/10.5683/SP3/FOJDLJ) by researchers who meet the criteria for access to confidential data.].

Reviewers' comments:

Reviewer's Responses to Questions

**Comments to the Author**

1. Is the manuscript technically sound, and do the data support the conclusions?

Reviewer #1: Yes

Reviewer #2: Partly

2. Has the statistical analysis been performed appropriately and rigorously?

Reviewer #1: N/A

Reviewer #2: N/A

3. Have the authors made all data underlying the findings in their manuscript fully available?

Reviewer #1: Yes

Reviewer #2: No

4. Is the manuscript presented in an intelligible fashion and written in standard English?

Reviewer #1: Yes

Reviewer #2: Yes

Reviewer #1: This manuscript was a revised version of already assessed by 3 reviewers.

The authors certainly reply effectively to all comments provided in the appendix

my only comments are as follow:

1) typo line 43 - recommendation with a s

2) line 79 - The authors: can they clarify if it was another

team a sit is difficulty to understand who they are

3) line 95 and 98

please change the word things as it is very vague and do not bring any value or understanding of the useful content

4) line 150 add (BC)

5) line 525 - change 344 to 34

6) great aspect: the authors use evidence based material from previous published studies

7) rigorous method for data analysis

Reviewer #2: The manuscript describes the experiences of older adults and people with disabilities during the first few months of the COVID-19 pandemic. The authors describe Sen’s Capability approach and how it was used in their qualitative thematic analysis. There are some very impactful quotes and discussion points, however significant revisions are needed.

Major comments-

There is poor connection as to why these 4 populations were pooled in this publication, as there is little justification as to why these groups are being considered together. How do older adults without disability relate to people with disabilities? It seems like very disparate data and OA is being conflated with disability without justification as to why. This is an issue from the introduction and continues with little linkage or explanation between OA and the other groups in the results and discussion.

For the population group of other disabilities, was there some recruitment strategy for individuals with physical, cognitive, or other disabilities? Strongly suggest clarifying what types of disabilities are/could be includes here would add great value. The disability community is very heterogenous and some description would allow for a better understanding of capabilities/inequities this group experience that are distinct from the stroke or SCI groups. In results there is some of this description attached to quotes but there needs to be a cohesive description earlier. This also ties into the previous comment about the lack of connection between study groups.

The addition of Mitra, Trani et al, etc. based on other reviewer comments does not add value as currently written. As there is no discussion of how/why the authors consider this in their study (line 134). Perhaps this could be further explored in methods (lines 206-20) where capability approach is mentioned (but its use could also be further detailed).

Overall, a careful round of editing is needed as there are syntax and punctuation issues, some noted below.

Minor comments-

General: In some cases oxford comma is used and other times is not, suggest being consistent.

Introduction: lines 57-59 read as though people with disabilities wish to receive higher priority. Whereas their concern lies more with medical ableism and not being fairly assessed based on quality of life bias. Suggest re-phrasing to emphasize the validity of their concern.

Lines 69-71 needs editing to clarify.

Line 98 check comma placement between “broader” and “social”.

Line 109 “lives in place” suggest change to “lives in a place”

Lines 120-123 presents Mirta’s model with and without capitalizations.

Methods: Line 142 used involved twice, would suggest other word choice.

Line 145 suggest adding ethic review #s

Lines 162,166 repeats the same “The majority of participants were born in Canada”.

Lines 189-190 were the interviewers also the coders? IE. Did people use their lived/living experience of disability to inform the analysis? You reference positionality of interviewers but it is not clear for the rest of the research process (lines 211-213).

Lines 189-207 Needs clarification if there were different codes for each of the 4 populations (IE 4 different coding guides were made) or 1 coding book based on all 4 populations. A flowchart to illustrate this process could be a good addition.

This may help you for describing processes of qualitative data integration as well as how you used Sen’s framework in your methods: DiMartino, L., Carroll, A.J., Ridgeway, J.L. et al. Development of a method for qualitative data integration to advance implementation science within research consortia. Implement Sci Commun 6, 21 (2025). https://doi.org/10.1186/s43058-025-00701-4

Results:

Line 216-220 punctuation and syntax issues, including multiple colons in one sentence.

Line 244 missing period

Line 342 “the challenges the experienced” should be “the challenges they experiences”

Line 353 “a physically health life” should be “healthy”

Line 374 In quote - Girl Guide lades – would this not be ladies? Check for typo or note sic?

Line 457-462 Quote about appreciation of healthcare workers is not clear how it links to the sub-theme of “wanting to exert control”

Discussion:

Line 524 citation # not correct

Line 527 “increased disability” should be changed. This is equating health with disability, which does not align with how you describe disability in lines 64-65 as health consequences or how you describe it in the Human Development Model.

Line 547 – “without abilities” should be disabilities?

Limitations:

Line 551- You describe COVID-19 as an epidemic here, but pandemic elsewhere.

**Do you want your identity to be public for this peer review?** For information about this choice, including consent withdrawal, please see our Privacy Policy

Reviewer #1: No

Reviewer #2: No

---

## [Author Response · Author response to Decision Letter 1]

30 Apr 2025

Please see responses in the included Response to Reviewers documents.

---

## [Editor Report · Decision Letter 1]

12 May 2025

Pandemic-related experiences of older adults and people with disabilities

PONE-D-24-34362R1

Dear Dr. Miller,

We’re pleased to inform you that your manuscript has been judged scientifically suitable for publication and will be formally accepted for publication once it meets all outstanding technical requirements.

Kind regards,

Maria Berghs, PhD

Academic Editor

PLOS ONE
---

## [Editor Report · Acceptance letter]

PONE-D-24-34362R1

PLOS ONE

Dear Dr. Miller,

I'm pleased to inform you that your manuscript has been deemed suitable for publication in PLOS ONE. Congratulations! Your manuscript is now being handed over to our production team.

Kind regards,

on behalf of

Dr. Maria Berghs

Academic Editor

PLOS ONE